# Establishment and Validation of a GC–MS/MS Method for the Quantification of Penicillin G Residues in Poultry Eggs

**DOI:** 10.3390/foods10112735

**Published:** 2021-11-09

**Authors:** Chujun Liu, Yawen Guo, Bo Wang, Lan Chen, Kaizhou Xie, Chenggen Yang

**Affiliations:** 1College of Animal Science and Technology, Yangzhou University, Yangzhou 225009, China; liuchujuncjl@163.com (C.L.); dx120200135@yzu.edu.cn (Y.G.); 2Joint International Research Laboratory of Agriculture & Agri-Product Safety, Yangzhou University, Yangzhou 225009, China; dz120180009@yzu.edu.cn (B.W.); chenlan9326@163.com (L.C.); 3College of Veterinary Medicine, Yangzhou University, Yangzhou 225009, China; 4College of Chemistry & Chemical Engineering, Yangzhou University, Yangzhou 225009, China; yangcg@yzu.edu.cn

**Keywords:** penicillin G, GC–MS/MS, ASE, poultry eggs, SPE

## Abstract

A simple and sensitive gas chromatography–tandem mass spectrometry (GC–MS/MS) method was established for the quantitative screening of penicillin G residues in chicken and duck eggs (whole egg, yolk and albumen). The analyte was separated on a TG-1MS capillary column (30.0 m × 0.25 mm i.d., 0.25 μm) with an external calibration method and electron impact (EI) ionization. Samples were pretreated using an accelerated solvent extraction (ASE) procedure followed by solid-phase extraction (SPE) on HLB cartridges (60 mg/3 mL). The derivative, which was safer and easier to store than penicillin G, was obtained by reacting trimethylsilyl diazomethane (TMSD) with penicillin G. The method was validated by the following parameters: linearity, accuracy, precision, limit of detection (LOD) and limit of quantification (LOQ). The matrix-matched calibration curves had good linearity (R2 ≥ 0.9994) within the concentration range of LOQ–200.0 µg/kg for penicillin G in the sample matrices. In the same concentration range, the accuracy, in terms of recovery, was 80.31–94.50%; the relative standard deviation (RSD), intra-day RSD and inter-day RSD ranged from 1.24 to 3.44%, 2.13 to 4.82% and 2.74 to 6.13%, respectively. The LODs and LOQs of penicillin G in the matrices were in the ranges of 1.70–3.20 and 6.10–8.50 μg/kg, respectively. The applicability of the GC–MS/MS method was demonstrated by the determination of poultry eggs obtained from local markets with no penicillin G residues.

## 1. Introduction

Poultry eggs, mainly chicken and duck eggs, have always been essential products in the poultry industry. Poultry eggs are an alternative for meat because they are of low cost and high nutrition and are abundant in protein, fat, vitamins and inorganic salts. In addition, amino acids are the main component of protein in poultry eggs [1] and are essential and necessary substances for human life. The fat in poultry eggs is easily digested and absorbed in the human gastrointestinal tract with ample unsaturated fatty acids [1]. Thus, the majority of consumers are increasingly purchasing poultry eggs in the marketplace. To enhance the growth efficiency and well-being of poultry, antibiotics are widely used to prevent and treat disease.

Penicillin G, a naturally existing narrow-spectrum antibiotic, is commonly used to prevent diseases in poultry farming. This antibiotic is mainly used to control avian influenza and treat poultry oophoritis. Moreover, penicillin G could also prevent chronic respiratory disease and nonspecific infectious enteritis [2]. The proper and legitimate use of penicillin G could stimulate growth and improve feed efficiency in livestock and poultry. However, penicillin G has been abused for high profits in breeding by certain enterprises and individual farmers, which would directly lead to veterinary drug residues in livestock and poultry products [2]. Drug residues were shown to exist in poultry eggs when feed was overly or mistakenly supplemented with antibiotics or when laying poultry were given medicated feed. Additionally, poultry eggs containing drug residues are digested and absorbed by humans, which can cause harm, including itching skin or even death caused by allergic reactions, and pollution of the environment [3]. Considering the violation between the physicochemical properties of penicillin G and the physiological characteristics of poultry eggs, the use of penicillin G is usually prohibited by setting a maximum residue limit (MRL). To determine penicillin G in poultry eggs rapidly and sensitively, it is necessary to establish a method that poses high efficiency and shortens the time and cost.

In recent decades, several essential methods have been applied for the detection of penicillin G, such as thin-layer chromatography (TLC) [4], immunoassays [5], gas chromatography–mass spectrometry (GC–MS) [6] and liquid chromatography–tandem mass spectrometry (LC–MS/MS) [7]. Furthermore, compared to GC–MS and LC–MS/MS, TLC is relatively simple and inexpensive, but it is not very sensitive and efficient and cannot be used to analyze complicated substances [4]. GC–MS and LC–MS/MS separate penicillin G from sophisticated sample matrices due to the high resolution of the column, and both could precisely obtain the target for quantitation [6,7]. Additionally, the LC–MS/MS method also have the ability for qualification to detect the analyte. However, based on the nonvolatile nature of penicillin G, it should be derivatized with diazomethane when using GC–MS, which requires appropriate solutions for storing derivatives and is time-consuming [6]. Therefore, selecting a proper method is necessary to determine penicillin G in poultry eggs that could acquire high recoveries and exhibit high sensitivity. Gas chromatography–tandem mass spectrometry (GC–MS/MS) presents the ability of accurate qualification and quantitation. This technique has not been previously reported for the detection of penicillin G residues in poultry eggs. Furthermore, GC–MS/MS also has a high capability to resist matrix interference, with relatively high selectivity and specificity, which could quickly separate analytes from impurities. Moreover, MS/MS could exactly identify the structure of the target compounds by analyzing the mass-to-charge ratio of the separated ions obtained from the mass spectrum. Consequently, the GC–MS/MS technique is suitable for detecting trace drugs in complicated matrices through improved sensitivity.

To remove interfering substances in matrices, a simple and efficient sample preparation procedure was developed. Commonly, classical sample pretreatment techniques for the majority of edible tissues consists of extraction, purification, concentration and reconstitution [8,9]. The derivatization process is also included in the pretreatment in terms of physicochemical properties of hardly volatile penicillin G. To date, traditional liquid–liquid extraction (LLE) has been gradually substituted by the modern technique of accelerated solvent extraction (ASE) because of the relatively high complexity and low extraction efficiency of LLE [10]. Furthermore, the greatest advantages of ASE are time savings and solvent savings, as ASE utilizes elevated temperatures and pressures; the former could rapidly strengthen the ability of the extraction solutions to dissolve the target substance and the latter could keep the extraction solutions in a liquid state [11]. Hence, ASE exhibits superior extraction efficiency. Solid-phase extraction (SPE) is usually adopted for purification because of the significant features of strong specificity and sensitivity [12,13]. Finally, ASE and SPE were both chosen in this experiment. Moreover, the unstable diazomethane derivatizing reagent was replaced by trimethylsilyl diazomethane (TMSD), which is safer and more suitable for derivatization in this study [14,15]. In addition, precolumn derivatization was used.

The main purpose of this research was to establish and validate a qualitative and quantitative method for the detection and confirmation of penicillin G residues in poultry eggs by GC–MS/MS. At the same time, sample pretreatment parameters of the ASE and SPE procedure and derivatization reagent were optimized. Appropriate sample pretreatment not only promoted the accuracy of the quantitative results but also contributed to the maintenance and lengthening of the instrument life. Furthermore, the validated method could provide a scientific foundation for the determination of penicillin G residues in poultry eggs.

## 2. Materials and Methods

### 2.1. Chemicals, Reagents and Apparatus

The penicillin G potassium salt standard (PENG, purity ≥ 98.0%) was purchased from Sigma-Aldrich (St. Louis, MO, USA). HPLC-grade acetonitrile and methanol were both supplied by Merck Co., Ltd. (Huntertown County, NJ, USA). HPLC-grade ethanol was obtained from Fisher Scientific International Inc. (Pittsburgh, PA, USA). Analytical-grade potassium dihydrogen phosphate, n-hexane, sodium hydroxide and disodium hydrogen phosphate dodecahydrate were all provided by Sinopharm Chemical Reagent Co., Ltd. (Shanghai, China). TMSD (dissolved in 1.5 M hexane) was provided by Aladdin Reagent Co., Ltd. (Shanghai, China).

C18 cartridges (500 mg/6 mL) and differently sized Oasis HLB cartridges (200 mg/6 mL, 500 mg/6 mL and 60 mg/3 mL) were supplied by Bonaageer Technology Co., Ltd. (Tianjin, China) and Waters Corporation (Milford, MA, USA), respectively. The capillary columns used for separation were 30 m × 0.25 mm (inside diameter), 0.25 μm TG-1MS and TG-5MS columns. Infusorial earth, 28 mm extraction cell filters, and ultrapure water provided by the Smart2-Pure system were supplied by Thermo Fisher Scientific Co., Ltd. (Shanghai, China). In addition, 0.22 μm filter membranes were provided by Anpu Technology Co., Ltd. (Shanghai, China). The resistivity of the ultrapure water at room temperature was 18.2 MΩ cm^–1^, which met the national laboratory water requirements (GB6682-1992). A TBOYS fully automatic multitube vortex oscillator and an N-EVAP-112 nitrogen blowing instrument were purchased from Troemner Co., Ltd. (Thorofare, NJ, USA) and Organomation Associates, Inc. (Shanghai, China), respectively. A P300H ultrasonic cleaner, a 5810R desktop high-speed refrigerated centrifuge and an FD115 oven were obtained from Elma (Shanghai, China), Eppendorf (Hamburg, Germany) and Binder (Tuttlingen, Germany), respectively. An ASE 350 instrument was provided by Thermo Fisher (Waltham, MA, USA).

### 2.2. GC–MS/MS Conditions

The screening of the target analyte was implemented on a Trace 1300 gas chromatograph equipped with a Triplus RSH autosampler and a TSQ 8000 triple quadrupole tandem mass spectrometer (Thermo Fisher Scientific Co., Ltd., Waltham, MA, USA). TG-1MS, a fully nonpolar capillary column (30.0 m × 0.25 mm i.d., 0.25 μm) consisting of 100% dimethyl polysiloxane, was used for the separation of the target analyte. Helium with a purity above 99.999% was used as the carrier gas at a constant flowrate of 1.0 mL/min and pressure of 60 psi. Injection was implemented in splitless mode over 1.0 min, and the injection volume was 1.0 μL. The shunt flow was set at 50.0 mL/min. The carrier gas saving time and flow were 2.0 min and 20.0 mL/min, respectively. The inlet temperature was held at 280 °C. The optimized initial temperature of the oven was set at 100 °C for 1 min, increased at a rate of 30 °C per minute to 220 °C for 1 min, and then ramped at the same rate until reaching 280 °C for 5 min.

For the parameters of mass spectrometry, electron impact (EI) ionization was implemented with an energy of 70 eV. Argon was used as the collision gas at 40 psi, with a high purity of 99.999%. The MS transmission line and ion source temperature were both 280 °C. In addition, the dwell time was set at 5 min. Data were acquired in full scan and selected reaction monitoring (Auto SRM) modes. The mass spectra of the derivative produced by penicillin G and TMSD were acquired through the above scanning modes, as shown in Figure 1. Fragment ions with the characteristics of a high mass-to-charge ratio and the most abundant precursor ion were selected for qualitative analysis, and the product ions were selected at the optimum collision energy for quantitative detection. Finally, the monitoring ion pair was obtained. The retention time and MS parameters are shown in Table 1.

### 2.3. Preparation of Standard Stock and Working Solutions

The standard stock solution of 1.0 mg/mL penicillin G was configured by dissolving approximately 10.20 mg of the standard with ethanol into a 10-mL brown volumetric flask and then storing it in a −70 °C freezer for up to five months. The standard working solutions were freshly prepared with ethanol at the appropriate concentrations (100.0 μg/mL, 10.0 μg/mL, 1.0 μg/mL and 100.0 ng/mL) and were used for the preparation of spiked samples and matrix-matched standards. The working solutions were stable for one month at 4 °C.

### 2.4. Sample Pretreatment

Analyte-free poultry eggs (chicken eggs and duck eggs) were produced by forty laying hens and laying ducks from Haiyang yellow chicken (Jiangsu Jinghai Poultry Industry Group Co., Ltd., Nantong, China) and Gaoyou duck (Jiangsu Gaoyou Duck Company, Yangzhou, China) during a period of 28–30 weeks; the poultry were given complete feed containing no veterinary drugs. The chicken eggs and duck eggs, as blank samples, were homogeneously divided into whole eggs, yolk and albumen and stored in a freezer at −34 °C.

The blank samples were thawed at 25 °C prior to use. Two grams of homogenized blank samples were weighed precisely in a mortar, spiked with the appropriate amount of diatomite, ground, and transferred to a 22 mL extraction cell. Additional diatomite was used to fill the space of the extraction cell when it was not completely full, and the sample was then extracted on the mechanical arm of ASE 350. The extraction procedure mainly included two steps. First, n-hexane was used to remove the fat from the matrices since it would affect the extraction efficiency. Second, 80% acetonitrile was selected for extracting the target analytes from the poultry eggs. The main parameters were similar except for the extraction times. The number of extraction cycles was once in the first step and twice in the second step. The following parameters were the same in both steps: extraction pressure, 1500 psi; extraction time, 5 min; extraction temperature, 30 °C; and nitrogen purging time, 60 s. The automatic washing times was once between each sample, and the volume of the total amount of washing solvent was 40%. Finally, the extract was collected in centrifuge tubes for use. Then, HLB SPE cartridges (60 mg/3 mL) were used for purification and were preconditioned with 3 mL of methanol (water and phosphate buffer) for activation and equilibration. The extractant was passed through the cartridge at a constant flow rate of 2.0 mL/min. Next, to separate the impurities from the matrices, the cartridge was washed with 3 mL of phosphate buffer and 3 mL of water 1 time and then dried under full vacuum for 2 min. The target analyte was eluted with 3 mL of 1% methanol in acetonitrile twice, and then the eluate was collected in 10 mL centrifuge tubes for drying in a nitrogen blowing apparatus at 35 °C. Subsequently, 100 μL of ethanol was utilized to redissolve the sample containing the analyte and was vortexed homogeneously for 1 min. Then, the reconstituted mixture was derivatized with 400 μL of TMSD for 30 min under a 30 °C oven in the dark. Afterward, the derivative solution was brought to 1 mL with ethanol in a 2.0 mL centrifuge tube. After vortexing briefly for 1 min and centrifuging for 10 min at 12,000× *g*, the final target solutions were filtered through 0.22 μm organic-phase syringe filters, and 1 μL of the analytes was injected into the GC–MS/MS apparatus for analysis.

### 2.5. Validation of the Analytical Method

The GC–MS/MS method was validated in reference to the EU Commission regulation (EU) No. 37/2010 guidelines [16,17,18]. The evaluation parameters mainly included linearity, accuracy, limit of detection (LOD) and quantification (LOQ).

#### 2.5.1. Linearity

The linearity was confirmed by constructing a matrix-matched calibration curve at spiked concentrations of the LOQ and 10.0, 25.0, 50.0, 100.0, 150.0 and 200.0 µg/kg for penicillin G. Afterward, the blank matrix extracts were prepared with 10 blank chicken eggs and duck eggs free of drug residues that were subjected to the above sample pretreatment procedures and then stored in a freezer at −34 °C. The matrix-matched calibration curve for quantitation was generated with the added concentration of penicillin G standard working solutions in blank samples as the x axis and the peak area of the quantitative ion pair *m*/*z* 174.1 > 114.1 * of the derivatives as the y axis. Six replicates of each sample spiked with the standard working solutions were analyzed by GC–MS/MS.

#### 2.5.2. Accuracy and Precision

The accuracy and precision of the GC–MS/MS method were based on recovery and repeatability. The intra-day precision and inter-day precision were measured by analyzing six replicates with added concentrations equal to the LOQ, 0.5 MRL, 1.0 MRL and 2.0 MRL for each sample. In addition, the intra-day precision was determined for the GC–MS/MS instrument and matrix-matched calibration curve on one day at different time points; the inter-day precision was determined by the same GC–MS/MS instrument using different matrix-matched calibration curves within one week and on different days. The recovery was estimated at four concentrations (LOQ, 0.5 MRL, 1.0 MRL and 2.0 MRL) with six replicates of penicillin G in the matrices that were analyzed by GC–MS/MS. Finally, the spiking recovery was calculated by obtaining the concentration from the detection results in the blank matrix-matched calibration curves.

#### 2.5.3. LOD and LOQ

The LOD and LOQ were also confirmed as the penicillin G concentration corresponding to a signal-to-noise (S/N) ratio of 3 and 10, respectively. Consequently, the mean S/N ratio was calculated with six replicates in every sample analysis and determined by the established GC–MS/MS method. Additionally, the LOQ could be confirmed in the method when the recovery was above 70% and the accuracy was less than 20% [19,20,21].

## 3. Results and Discussion

### 3.1. Selection of Solvents

The appropriate selection of solvents can not only directly affect the stability of the analytes but also have a significant influence on the results. Penicillin G, with strong polarity, can dissolve in water, methanol, ethanol and acetonitrile [22]. This experiment selected water, methanol, ethanol and different proportions (70%, 80% and 90%) of acetonitrile as the solvents to dissolve the penicillin G standard. The results showed that the penicillin G aqueous solution needed to be used immediately after preparation, as it is easily degraded at room temperature and was difficult to store at 4 °C. Moreover, the derivatization reaction between penicillin G and TMSD did not proceed in an aqueous solution. This occurred mainly because TMSD is not soluble in water [15]. Different ratios of acetonitrile could dissolve the penicillin G standard, and the dissolution rate obtained with 80% acetonitrile was the highest. When methanol and ethanol were chosen as the solvents, both could produce the derivative with good peaks. However, the derivative produced by TMSD and penicillin G dissolved in ethanol was more stable than that obtained with methanol. Additionally, compared with 80% acetonitrile, ethanol was the better choice due to the faster dissolution rate and higher peak area of the derivative. Therefore, ethanol was selected as the solvent for penicillin G dissolution.

### 3.2. Optimization of Sample Pretreatment

In this study, the optimization of the sample pretreatment process was mainly conducted by the selection of extraction, purification and derivatization methods. In this study, the extraction methods mainly adopted liquid–liquid extraction and ASE extraction. In the process of liquid–liquid extraction, 2.0 ± 0.02 g of a homogeneous blank sample was accurately weighed, 10 mL of 0.2 M phosphate buffer solution (pH 8.0) was added, and the sample was mixed for 5 min in an automatic multi-tube vortex mixer at a rotating speed of 2000× *g* and sonicated for 5 min. The sample matrices were centrifuged at a high speed of 10,000 r/min at 4 °C for 10 min. The extraction procedure was repeated twice. Then, the supernatants were combined and transferred to a 50-mL polypropylene centrifuge tube. A total of 5 mL of n-hexane was added once for degreasing, and the vortex mixing procedure and centrifugation conditions were the same as above. Finally, the sample matrices containing analytes were collected. Then, the LLE and ASE methods and solvents for the extraction of analytes were compared. First, as shown in Table 2, for the extraction of penicillin G in poultry eggs, the ASE procedure obtained a recovery of 81%, higher than that of the LLE method, as ASE can sufficiently extract the determined components in matrices by reducing the deviations caused by human error. Furthermore, the extractants, for instance, a 0.2 M phosphate buffer solution and different proportions of acetonitrile (60%, 70%, 80%, 90% and 100%), were investigated for the extraction efficiency of the analytes in poultry eggs. Acetonitrile had a better extraction efficiency than the 0.2 M phosphate buffer solution in terms of recovery. Moreover, 80% acetonitrile was the optimum extractant, with a high recovery of 83%, as shown in Table 2. Thus, 80% acetonitrile and the ASE method were chosen for the subsequent extraction of analytes in samples.

Penicillin G, a β-lactam antibiotic with high polarity, requires an SPE cartridge with broad selectivity [22]. SPE is commonly used in the process of purification and can reduce the effects of ion enhancement caused by components in poultry eggs [23]. C18 and HLB SPE cartridges are typically used for the preconcentration of polar analytes. This experiment compared C18 cartridges (500 mg/6 mL) and different specifications of Oasis HLB cartridges (200 mg/6 mL, 500 mg/6 mL and 60 mg/3 mL). The results showed that the recoveries were the highest when 60 mg/3 mL HLB cartridges were utilized, and the recoveries were below 50% when 500 mg/6 mL and 200 mg/6 mL HLB cartridges were implemented; moreover, there were impurity peaks in the mass spectrum when C18 cartridges were used. Therefore, 60 mg/3 mL Oasis HLB cartridges were selected.

The proper selection of the derivatization reagent is crucial in GC–MS/MS analysis. Diazomethane was once proposed to derivatize penicillin G with an internal calibration method [6]. Moreover, the derivatization reaction could be rapidly completed under the conditions of a neutral environment and only produced N2. However, diazomethane has the following drawbacks: it is difficult to prepare and easily decomposes at room temperature. Diazomethane also has high toxicity and thermal instability and is explosive, and impurity peaks appear if it is not used immediately [24]. Taking all of this into consideration, TMSD, which is more stable and easier to use than diazomethane, was chosen as an alternative in this experiment and overcame the shortcomings of diazomethane [24]. Hence, the esterification reaction of penicillin G and TMSD was chosen for derivatization, producing penicillin G trimethylsilyl methyl ester as the derivative. Moreover, the conditions (TMSD volume, temperature and time) of the derivatization reaction were optimized. Figure 2 shows the results for derivatization. The optimum derivatization reaction conditions were as follows: TMSD volume, 400 µL; temperature, 30 °C; and time, 30 min. These parameters were optimized in terms of the peak area of the derivative. Additionally, the stability of the derivative was confirmed for the validation of reproducibility. As shown in Figure 2, the derivative stably existed during a whole day. On the other hand, the external calibration method was used to reduce the analytical cost, and the results were good. Thus, TMSD in combination with the external calibration method was selected in this study.

### 3.3. Method Validation

#### 3.3.1. Linearity

Matrix effects (MEs) are defined as the enhancement or suppression of the analyte response in samples due to components other than the analytes, and MEs can affect the accuracy [25]. Certain factors, such as the kind of analyte, the properties of the samples and the instrument detection method, can modify MEs [25]. The enhancement of MEs commonly occurs in GC–MS/MS analysis. MEs can reduce the loss of analyte components, increasing the response, and are usually produced in the inlet or column [26]. Thus, to avoid and diminish the effects of MEs, several methods, such as the establishment of appropriate instrument methods, the optimization of sample pretreatment and the development of matrix-matched calibration curves, can be applied [25]. In this experiment, MEs were reduced by optimizing the sample pretreatment and establishing matrix-matched calibration curves, and good results were obtained. The matrix-matched calibration curves mainly exhibited linearity. The linearity of standard working solutions with various added concentrations of the analyte compound versus peak area was validated in poultry eggs over the range of LOQ-200.0 µg/kg. The determination coefficient (R2) values obtained by the external calibration method were higher than 0.9994 for penicillin G in the sample matrices, as shown in Table 3. Compared with the internal standard method, the external calibration method is less expensive [6]. Moreover, the calibration curves obtained with the external calibration method exhibited better linearity than those obtained with the internal standard method [6].

#### 3.3.2. Accuracy and Precision

Accuracy and precision can directly reflect the determination of penicillin G in poultry eggs by the GC–MS/MS method. Accuracy was calculated according to the recovery, which was calculated by the ratio between the measured value and true value; precision was calculated according to the relative standard deviation (RSD), which represented the repeatability of the results. Recovery was regulated by the 2002/657/EC resolution of the European Union within the range of 70–110% [19]. Additionally, the method became more and more reliable with the increasing recoveries and decreasing RSDs. In this research, the recovery and precision were obtained by adding concentrations of the LOQ and 25.0, 50.0 and 100.0 µg/kg penicillin G to poultry eggs. In poultry eggs spiked with penicillin G, the measured recovery was within the range of 80.31–94.50%; the RSD, intra-day RSD and inter-day RSD ranged from 1.24 to 3.44%, 2.13 to 4.82% and 2.74 to 6.13%, respectively. The results are summarized in Table 4. In addition, the total ion chromatograms (TICs) and chromatograms of the quantitative and qualitative ions for the analytes in the blank matrices and samples spiked with the 25.0 μg/kg standard are shown in Figure 3A,B. All peaks were well separated with the retention time of 10.85 min.

#### 3.3.3. LODs, LOQs and Sensitivity

The LOD and LOQ in a method are crucial for accurate qualification and quantitation of the analytes. Moreover, high sensitivity can be determined by low LODs and LOQs [27]. As shown in Table 4, the LOD of penicillin G ranged from 1.70 to 3.20 μg/kg in poultry eggs, and the LOQ of penicillin G ranged from 6.10 to 8.50 μg/kg in poultry eggs.

### 3.4. Application of the Method

The GC–MS/MS analytical method was applied for the determination of penicillin G in 50 poultry eggs purchased from local supermarkets. There were no penicillin G residues detected in the tested samples. Therefore, the GC–MS/MS method could be adopted for the qualification and quantitation of penicillin G residues in poultry eggs.

## 4. Conclusions

This study proposed the development and validation of an inexpensive and sensitive GC–MS/MS method for the detection of penicillin G in poultry eggs (whole eggs, yolk and albumen). At the same time, the sample pretreatment method was optimized, including the selection and optimization of the ASE parameters (temperature, time and flushing percentage) and the derivatization reaction conditions, improving the repeatability and extraction efficiency of the analyte. Additionally, the ASE procedure reduced the effects of the MEs and saved reagents. TMSD was selected as the derivatization reagent, allowing for relatively safe and rapid derivatization. This method provides a new analytical technique with high recovery and sensitivity for the determination of penicillin G in poultry eggs.

## Figures and Tables

**Figure 1 foods-10-02735-f001:**
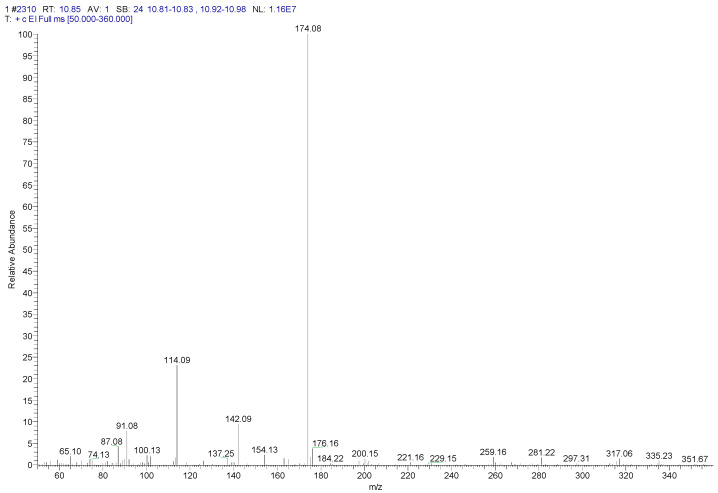
Mass spectrogram of penicillin G derivatives.

**Figure 2 foods-10-02735-f002:**
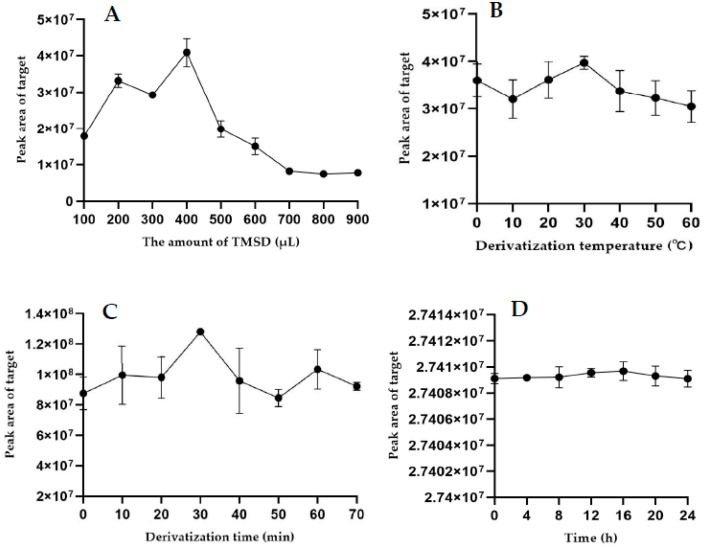
Effect of the amount of TMSD (**A**), derivatization temperature (**B**), and derivatization time (**C**) on the derivatization reaction and the stability of the penicillin G derivative within 24 h (**D**).

**Figure 3 foods-10-02735-f003:**
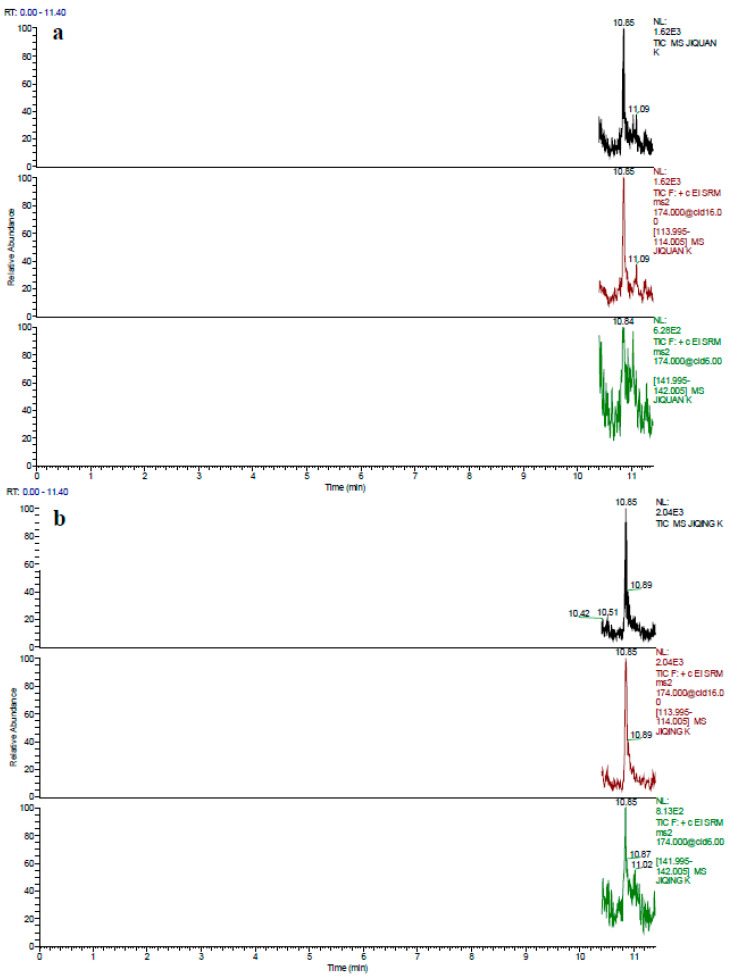
(**A**) Total ion chromatograms (TICs) and quantitative and qualitative ion chromatograms of blank chicken whole egg (**a**), albumen (**b**) and yolk (**c**), and duck whole eggs (**d**), albumen (**e**) and yolk (**f**). (**B**) Total ion chromatograms (TICs) and quantitative and qualitative ion chromatograms of blank chicken whole egg (**a**), albumen (**b**) and yolk (**c**), and duck whole egg (**d**), albumen (**e**) and yolk (**f**), spiked with 25.0 μg/kg penicillin G.

**Table 1 foods-10-02735-t001:** Retention time and relevant MS parameters of penicillin G trimethylsilyl methyl ester.

Target Compound	Molecular Weight	Retention Time	Mass Transitions	Collision Energy
Penicillin G trimethylsilyl methyl ester	421.48	10.85	174.1 > 91.1	6
			174.1 > 114.1 *	16
			174.1 > 142.1	6

Note: * Quantitative ion pair.

**Table 2 foods-10-02735-t002:** Effect of extraction method and different extraction reagents on the recovery of penicillin G in poultry eggs (%) (*n* = 6).

Matrix	Extraction Method		Extraction Reagent				
	LLE	ASE	60% Acetonitrile	70% Acetonitrile	80% Acetonitrile	90% Acetonitrile	100% Acetonitrile
Chicken whole egg	71.06 ± 2.53	87.96 ± 1.54	59.16 ± 2.81	62.21 ± 3.54	84.27 ± 1.13	68.15 ± 2.76	79.66 ± 1.78
Chicken albumen	62.58 ± 1.62	85.32 ± 1.38	57.82 ± 3.01	65.86 ± 1.75	85.12 ± 1.59	64.32 ± 1.88	75.63 ± 2.19
Chicken yolk	73.95 ± 1.58	86.19 ± 2.13	55.68 ± 2.35	61.94 ± 2.26	86.74 ± 2.11	62.87 ± 1.64	73.24 ± 2.61
Duck whole egg	75.31 ± 2.39	84.56 ± 1.42	60.37 ± 3.12	63.56 ± 2.87	83.46 ± 2.05	65.92 ± 2.28	80.17 ± 2.38
Duck albumen	70.59 ± 2.73	81.74 ± 1.86	54.39 ± 2.46	60.32 ± 1.94	85.52 ± 1.61	63.47 ± 2.53	77.54 ± 1.96
Duck yolk	65.18 ± 2.21	89.28 ± 1.95	56.45 ± 1.82	61.78 ± 1.36	87.36 ± 1.84	62.51 ± 2.72	78.39 ± 3.07

**Table 3 foods-10-02735-t003:** Linear regression equations, determination coefficients and linearity ranges of penicillin G in poultry eggs (*n* = 6).

Matrix	Linear Regression Equation	Determination Coefficient (R2)	Linearity Range (µg/kg)
Chicken Whole egg	y = 24,281x + 56,797	0.9994	7.90–200.0
Chicken albumen	y = 17,204x − 32,177	0.9995	6.80–200.0
Chicken yolk	y = 26,719x − 45,245	0.9994	8.50–200.0
Duck whole egg	y = 32,395x − 19,611	0.9998	7.40–200.0
Duck albumen	y = 16,737x − 14,161	0.9996	6.10–200.0
Duck yolk	y = 29,856x − 69,373	0.9995	6.40–200.0

**Table 4 foods-10-02735-t004:** Recovery and precision of penicillin G added to blank poultry eggs (*n* = 6).

Tissues	Added Level (μg/kg)	Recovery (%)	RSD (%)	Intra-Day RSD (%)	Inter-Day RSD (%)	LOD (µg/kg)	LOQ (µg/kg)
Chicken whole egg	7.90	85.13 ± 2.08	2.44	4.12	5.09	2.50	7.90
	25.00	86.46 ± 1.51	1.75	2.13	4.28		
	50.00 *^α^*	88.72 ± 1.61	1.82	2.47	3.52		
	100.00	91.56 ± 1.87	2.04	3.45	4.58		
Chicken albumen	6.80	83.21 ± 2.17	2.61	4.82	5.25	2.10	6.80
	25.00	84.71 ± 1.45	1.71	3.26	4.36		
	50.00 *^α^*	93.11 ± 2.45	2.64	4.75	5.43		
	100.00	94.50 ± 1.18	1.24	3.21	3.85		
Chicken yolk	8.50	83.05 ± 1.43	1.73	4.63	4.97	3.20	8.50
	25.00	85.14 ± 2.38	2.80	3.15	4.43		
	50.00 *^α^*	86.23 ± 2.97	3.44	3.96	4.62		
	100.00	89.48 ± 3.01	3.36	4.27	5.18		
Duck whole egg	7.40	84.16 ± 1.39	1.65	4.08	4.56	2.40	7.40
	25.00	87.81 ± 2.24	2.55	3.61	4.23		
	50.00 *^α^*	90.16 ± 1.47	1.63	3.42	3.79		
	100.00	93.15 ± 2.12	2.28	3.35	4.38		
Duck albumen	6.10	80.31 ± 1.34	1.67	2.78	5.16	1.70	6.10
	25.00	81.67 ± 1.12	1.38	3.19	4.35		
	50.00 *^α^*	84.13 ± 2.14	2.54	2.31	3.18		
	100.00	92.16 ± 2.53	2.75	3.67	6.13		
Duck yolk	6.40	83.29 ± 1.56	1.87	3.31	2.74	1.80	6.40
	25.00	86.64 ± 2.26	2.61	2.91	3.23		
	50.00 *^α^*	87.56 ± 2.73	3.11	3.57	4.09		
	100.00	90.47 ± 1.85	2.05	4.25	4.83		

*^α^* Maximum residue limits.

## Data Availability

All data related to the research are presented in the article.

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
