# Peer review of "Establishment and Validation of a GC–MS/MS Method for the Quantification of Penicillin G Residues in Poultry Eggs"

_foods, 2021, doi:10.3390/foods10112735_

Round 1

Reviewer 1 Report

In the paper Establishment and validation of a GC–MS/MS method for the quantification of penicillin G residues in poultry eggs, authors report a metod to determine penicillin G residues in chicken and duck eggs by using ASE procedure followed by solid-phase extraction (SPE) on HLB cartridge bollowed a chromatography-tandem mass spectrometry (GC-MS/MS) method. 
Paper is not novely but interesting and some parts can be emproved.
I suggest major revision.

Introduction.
Line 90 
Solid-phase extraction (SPE) is usually adopted for purification because of the significant features of strong specificity and sensitivity [12]

Please add recent pubblicaton such as Presence and biodistribution of perfluorooctanoic acid (PFOA) in Paracentrotus lividus highlight its potential application for environmental biomonitoring. Scientific reports, 11(1), 1-8.

2.5. Validation of the Analytical Method 201
The GC-MS/MS method was validated in reference to the Commission regulation 202
(EU) No. 37/2010 guidelines [15]. The evaluation parameters mainly included linearity, accuracy, limit of detection (LOD) and quantification (LOQ). 
Please, add referecens. 
For examle use, ase reported in literature (Volatile profiles of emissions from different activities analyzed using canister samplers and gas chromatography-mass spectrometry (GC/MS) analysis: a case study. International journal of environmental research and public health, 2017, 14.2: 195.   and Photochemical sample treatment for extracts clean up in PCB analysis from sediments. Talanta, 2013, 103: 349-354.

Line 232
 Additionally, the 232
LOQ could be confirmed in the method when the recovery was above 70% and the accuracy was less than 20% [16]. 
Please report other references in order to confirm that these criteria are curently used. 
In this contex add [Hyphenated high performance liquid chromatography–tandem mass spectrometry techniques for the determination of perfluorinated alkylated substances in Lombardia region in Italy, profile levels and assessment: One year of monitoring activities during 2018. Separations, 2020, 7.1: 17.] and Photochemical sample treatment: A greener approach to chlorobenzene determination in sediments. Talanta, 2014, 129: 263-269.

Figure 2
Please add error bar 

Line 309 and Table 3
correlation coefficients (R2)
Please report R2 as Determination coefficient and use only 0.999 value

Author Response

Response to Reviewer 1 Comments

Point 1: Introduction.

Line 90

Solid-phase extraction (SPE) is usually adopted for purification because of the significant features of strong specificity and sensitivity [12]

Please add recent pubblicaton such as Presence and biodistribution of perfluorooctanoic acid (PFOA) in Paracentrotus lividus highlight its potential application for environmental biomonitoring. Scientific reports, 11(1), 1-8.

Response 1: I accept the suggestion and have added the publication “Presence and biodistribution of perfluorooctanoic acid (PFOA) in Paracentrotus lividus highlight its potential application for environmental biomonitoring”.

Point 2: 2.5. Validation of the Analytical Method 201

The GC-MS/MS method was validated in reference to the Commission regulation 202 (EU) No. 37/2010 guidelines [15]. The evaluation parameters mainly included linearity, accuracy, limit of detection (LOD) and quantification (LOQ).

Please, add referecens.

For examle use, ase reported in literature (Volatile profiles of emissions from different activities analyzed using canister samplers and gas chromatography-mass spectrometry (GC/MS) analysis: a case study. International journal of environmental research and public health, 2017, 14.2: 195. and Photochemical sample treatment for extracts clean up in PCB analysis from sediments. Talanta, 2013, 103: 349-354.

Response 2: I accept the advice and have added the references “Volatile profiles of emissions from different activities analyzed using canister samplers and gas chromatography-mass spectrometry (GC/MS) analysis: a case study” and “Photochemical sample treatment for extract clean up in PCB analysis from sediments” in line 202.

Point 3: Line 232

Additionally, the 232
LOQ could be confirmed in the method when the recovery was above 70% and the accuracy was less than 20% [16].
Please report other references in order to confirm that these criteria are curently used.
In this contex add [Hyphenated high performance liquid chromatography-tandem mass spectrometry techniques for the determination of perfluorinated alkylated substances in Lombardia region in Italy, profile levels and assessment: One year of monitoring activities during 2018. Separations, 2020, 7.1: 17.] and Photochemical sample treatment: A greener approach to chlorobenzene determination in sediments. Talanta, 2014, 129: 263-269.

Response 3: I accept the suggestion and have added the references “Hyphenated high performance liquid chromatography-tandem mass spectrometry techniques for the determination of perfluorinated alkylated substances in Lombardia region in Italy, profile levels and assessment: One year of monitoring activities during 2018” and “Photochemical sample treatment: A greener approach to chlorobenzene determination in sediments”.

Point 4: Figure 2

Please add error bar

Response 4: I accept the suggestion and have added an error bar in Figure 2.

Point 5: Line 309 and Table 3

correlation coefficients (R2)

Please report R2 as Determination coefficient and use only 0.999 value

Response 5: I accept the suggestion and have reported R2 as the determination coefficient in line 331 and table 3.

Reviewer 2 Report

Abstract: describe the acronym ASE.

Page 3, lines 146-147: "selective reaction monitoring (Auto SRM)" replace with "selected reaction monitoring (Auto SRM)".

Delete Table 5 and include the LOD and LOQ values in Table 4.

Figure 3A, include only chromatograms of one matrix (chicken whole egg/albumen/yolk or duck whole eggs/albumen/yolk).

Figure 3B, include only chromatograms of one matrix (chicken whole egg/albumen/yolk or duck whole eggs/albumen/yolk).

Section 3.5. Application of the method: How many samples was analysed? Describe the total number of samples.

Page 15, line 635: "Moreover, high sensitivity is usually determined by low LOD and LOQ". Are there any references for this statement?

I suggest removing section "3.4. Matrix effects", and include this discussion about matrix effects in section "3.3.1. Linearity".

Page 8, line 313: " than those obtained with an internal standard method". What was the internal standard used?

Section 2.5.3. LOD and LOQ: "The LOD and LOQ were considered the lowest concentrations for instrumental determination or quantification, respectively." I suggest deleting this sentence or then include a reference.

Section 3.2. Optimization of sample pretreatment: As described in section "3.1. Selection of solvents" the penicillin G standard was poorly soluble in 80% acetonitrile, however, high recoveries were obtained using 80% acetonitrile as extraction solvent... A discussion about this behaviour would be very interesting!

Table 2 describes results about LLE. Please, include a description of the LLE procedure used.

Author Response

Response to Reviewer 2 Comments

Point 1: Abstract: describe the acronym ASE.

Response 1: I accept the suggestion, and the acronym ASE has been defined as accelerated solvent extraction in the manuscript.

Point 2: Page 3, lines 146-147: "selective reaction monitoring (Auto SRM)" replace with "selected reaction monitoring (Auto SRM)".

Response 2: I accept the suggestion; "selective reaction monitoring (Auto SRM)" has been replaced with "selected reaction monitoring (Auto SRM)" on page 3, lines 146-147.

Point 3: Delete Table 5 and include the LOD and LOQ values in Table 4.

Response 3: I accept the suggestion and have deleted Table 5 and included the LOD and LOQ values in Table 4.

Point 4: Figure 3A, include only chromatograms of one matrix (chicken whole egg/albumen/yolk or duck whole eggs/albumen/yolk).

Response 4: In Figure 3A, what I want to express is that “the mass chromatograph of quantitative and qualitative ion” actually indicates the chromatograms of quantitative and qualitative ions. The mass spectrogram is shown in Figure 1. To be clear, “Total ion chromatograph (TIC) and mass chromatograph (MC) of quantitative and qualitative ion of blank chicken whole egg (a), albumen (b), and yolk (c) and duck whole eggs (d), albumen (e) and yolk (f)” has been replaced with “Total ion chromatograms (TICs) and quantitative and qualitative ion chromatograms of blank chicken whole egg (a), albumen (b), and yolk (c) and duck whole eggs (d), albumen (e) and yolk (f)”.

Point 5: Figure 3B, include only chromatograms of one matrix (chicken whole egg/albumen/yolk or duck whole eggs/albumen/yolk).

Response 5: “Total ion chromatograph (TIC) and mass chromatograph (MC) of quantitative and qualitative ion of blank chicken whole egg (a), albumen (b), and yolk (c) and duck whole egg (d), albumen (e) and yolk (f) spiked with 25.0 μg/kg penicillin G” has been replaced with “Total ion chromatograms (TICs) and quantitative and qualitative ion chromatograms of blank chicken whole egg (a), albumen (b), and yolk (c) and duck whole egg (d), albumen (e) and yolk (f) spiked with 25.0 μg/kg penicillin G”.

Point 6: Section 3.5. Application of the method: How many samples was analysed? Describe the total number of samples.

Response 6: A total of 50 samples were analyzed in section 3.4.

Point 7: Page 15, line 635: "Moreover, high sensitivity is usually determined by low LOD and LOQ". Are there any references for this statement?

Response 7: In Page 16, line 628, the reference “Considerations on the determination of the limit of detection and the limit of quantification in one-dimensional and comprehensive two-dimensional gas chromatography” supports the indicated statement.

Point 8: I suggest removing section "3.4. Matrix effects", and include this discussion about matrix effects in section "3.3.1. Linearity".

Response 8: I accept the suggestion and have removed section "3.4. Matrix effects"; the discussion of matrix effects has been moved to section "3.3.1. Linearity".

Point 9: Page 8, line 313: " than those obtained with an internal standard method". What was the internal standard used?

Response 9: In Page 9, lines 335-336, the internal standard used is that referenced in the method in “Isotope dilution GC-MS of benzylpenicillin residues in bovine muscle”.

Point 10: Section 2.5.3. LOD and LOQ: "The LOD and LOQ were considered the lowest concentrations for instrumental determination or quantification, respectively." I suggest deleting this sentence or then include a reference.

Response 10: I accept the suggestion and have deleted "The LOD and LOQ were considered the lowest concentrations for instrumental determination or quantification, respectively."

Point 11: Section 3.2. Optimization of sample pretreatment: As described in section "3.1. Selection of solvents" the penicillin G standard was poorly soluble in 80% acetonitrile, however, high recoveries were obtained using 80% acetonitrile as extraction solvent... A discussion about this behaviour would be very interesting!

Response 11: The experiment in section 3.1 in the manuscript was repeated, and it was found that 80% acetonitrile can dissolve the penicillin G standard.

Point 12: Table 2 describes results about LLE. Please, include a description of the LLE procedure used.

Response 12: I accept the suggestion and have described the LLE procedure in section 3.2.

Round 2

Reviewer 1 Report

All corrections were made.

Reviewer 2 Report

MS has been improved.